# The Effect of Functional Fiber on Microbiota Composition in Different Intestinal Segments of Obese Mice

**DOI:** 10.3390/ijms22126525

**Published:** 2021-06-18

**Authors:** Chuanhui Xu, Jianhua Liu, Jianwei Gao, Xiaoyu Wu, Chenbin Cui, Hongkui Wei, Jian Peng, Rong Zheng

**Affiliations:** 1Department of Animal Nutrition and Feed Science, College of Animal Science and Technology, Huazhong Agricultural University, Wuhan 430070, China; xuchuanhui001@webmail.hzau.edu.cn (C.X.); xbmzdxwxy@webmail.hzau.edu.cn (X.W.); ccb490464662@163.com (C.C.); weihongkui@mail.hzau.edu.cn (H.W.); 2Department of Animal Genetics and Breeding, College of Animal Science and Technology, Huazhong Agricultural University, Wuhan 430070, China; liujianhua@webmail.hzau.edu.cn (J.L.); JianweiG@webmail.hzau.edu.cn (J.G.); 3The Cooperative Innovation Centre for Sustainable Pig Production, Wuhan 430070, China

**Keywords:** microbiota composition, different segment, dietary fiber, obese mice, traceability analysis

## Abstract

The gastrointestinal tract is a heterogeneous ecosystem with distinct, stratified environments, which leads to different microbial composition in different intestinal segments. The regional heterogeneity of intestinal microbiota complicates the relationship between diet and microbiota. Few studies have focused on the effects of different diets on microbiota in different intestinal segments. This study aimed to investigate the effects of functional fiber on the microbial composition in multiple intestinal segments from a high-fat diet compared with a normal chow diet. We found that the response of microbiota from different intestinal segments to diet was related to the intestinal physiologic function and the physicochemical properties of dietary nutrients. A high-fat diet drove changes in the microbial composition in the hindgut, possibly by affecting the digestive environment of the foregut, and increased the regional heterogeneity of the whole intestinal microbiota. The supplementation of functional fiber promoted the microbial transfer and colonization from the anterior to the posterior intestinal segments, and increased the regional similarity of intestinal microbiota accordingly, particularly within the hindgut. The gut fermentation of the functional fiber, which mainly occurred in the hindgut, resulted in a significant change in the microbial composition and metabolism in the cecum and colon, with richer carbohydrate metabolism-related bacteria, including Mucispirillum, Prevotella, Anaerostipes, Oscillospira, Ruminococcus, Bacteroides, Coprococcus, Ruminococcus (Lachnospiraceae), and Allobaculum, and higher production of acetate and butyrate. We concluded that multiple regulatory mechanisms of diets which affect microbiota composition exist, including microbial metabolism, microbial migration, and the regulation of the intestinal environment.

## 1. Introduction

The mammalian intestine is divided into the duodenum, jejunum, ileum, cecum, colon, and rectum, each with different and distinct anatomies and functions [1]. In addition, the colonization of gut microorganisms differs between intestinal segments [2]. The microbial profiles of the fore-, mid-, and hind-intestine segments show a gradual increasing trend [3]. In terms of microbial composition, the foregut contains more facultative anaerobes, while the hindgut contains more strict anaerobes, and the relative abundance of Firmicutes gradually increases, while the Proteobacteria similarly decreases [2]. In terms of microbial function, the microbes in the foregut assist digestion and immune function, while those in the hindgut focus on fermented fiber [4,5].

There is evidence to suggest that obesity and related metabolic disorders may be caused by the disorder of intestinal microecology [6,7]. Previous studies have shown that high-fat diets do not cause obesity in germ-free mice; however, after microbial colonization, high-fat diets successfully induce obesity in mice [6]. Furthermore, fecal microbiota from genetically obese mice transplanted into germ-free mice could increase the fat storage capability of the recipient mice [7]. The gut microbiota mediate the fattening process of mice fed with a high-fat diet. However, the microbial composition of different intestinal segments and the microbial response to high-fat diets are not clear. It is also uncertain from which intestinal segment the microbiota begin to respond to the high-fat diet and promote obesity.

Dietary fiber is mainly used during microbial fermentation in the intestine to promote the production of SCFAs, which play an important role in maintaining intestinal health and barrier function [8]. Research indicates that dietary fiber could increase the relative abundance of Lactobacillus and Bifidobacterium, and promote the release of SCFAs, which may contribute to anti-obesity activity [9,10]. A new type of functional fiber (FF) was previously created by our research group, which plays a pivotal role in improving the intestinal barrier [11]. Rodent studies show that the successive supplementation of 4% or 6% FF in high-fat diets and normal chow diets improves obesity-related metabolic disorders, and regulates both the fecal microbiota composition and the production of SCFAs in a dose-dependent manner [12]. However, it is still unclear how FF affects the microbial composition and metabolism in the gut. Therefore, this study tracks the changes in intestinal microbiota along the axis of the gastrointestinal tract in different diets, illustrating how these diets can cause differences in the intestinal microbiota.

## 2. Results

### 2.1. α-Diversity

We undertook a comprehensive analysis of a total of 112 mouse intestinal (duodenum, ileum, cecum, colon) contents and fecal microbiomes by 16S rRNA sequencing. The average V3-V4 16S rRNA gene sequencing depth was 3,536,941 reads. After rarefying and removing contaminants, 4050 operational taxonomic units (OTUs) remained. The Venn diagram of common and unique OTUs in each group is shown in Figure 1. The number of common OTUs shared in the five different segments of the HF-C group was lower than that of the LF-C group, and the number of unique OTUs in each different intestinal segment was higher than that of the LF-C group. Compared with the HF-C group, the 4% and 6% FF groups contained a similar total OTU number, but increased common OTU numbers and decreased unique OTU numbers in the duodenum, ileum, and colon. Interestingly, the unique OTU numbers of cecum in the 4% and 6% FF groups were higher than those in the HF-C group. The number of observed species and Chao 1 index observed in the hindgut segment (cecum and colon) and fecal microbes were significantly higher than those of the foregut segment (duodenum and ileum) (*p* < 0.001) (Figure 2). These results indicated that there were differences in the composition of OTUs colonized in each intestinal segment between different groups.

### 2.2. β-Diversity

Principal component analysis showed the clusters of microbes in the foregut were separated from those in the hindgut and feces (Figure 3), which indicated a significant difference in the microbiota compositions between the foregut and hindgut (*p* = 0.001). The results of principal coordinates analysis (PCoA) on different intra-group intestinal segments showed that the microbial clusters of ileum, cecum, colon, and feces were significantly separated in the LF-C group; meanwhile, ileal microbiota were the most dispersed and colonic microbiota were more concentrated (*p* = 0.001) (Figure 4A). Microbial clusters of duodenum, ileum, and cecum were significantly separated in the HF-C group (*p* < 0.001) (Figure 4B). In the 4% and 6% FF groups, the microbial clusters of ileum, cecum, and colon were significantly separated (*p* < 0.001) (Figure 4C,D). The results of inter-group PCoA showed that the microbial clusters of duodenum, ileum, colon, and feces in the HF-C group were significantly separated from those in the LF-C group (Figure 5). The microbiota in each intestinal segment of the 4% and 6% FF groups were separated from the LF-C group, while the microbiota in the cecum, colon, and feces were separated from the HF-C group (Figure 5). These results indicated that the microbiota of the foregut segment (duodenum, ileum) and the hindgut segment (cecum, colon, and fecal) were significantly different. The hindgut segment had a higher microbial species diversity. A high-fat diet significantly affected the microbial composition of the whole intestine of the mice, and increased the differences in microbial composition between the foregut and the hindgut; functional fibers caused changes in the microbial composition of the hindgut.

### 2.3. Changes in Microbial Composition of Different Intestinal Segments

The composition of microbiota in different intestinal segments at the phylum and genus level is shown in Figure 6. At the phylum level, the dominant bacterial groups in all intestinal segments are Firmicutes and Bacteroides, followed by Proteobacteria and Actinomycetes, which account for at least 97.5% of the total number of microorganisms in all groups. At the genus level, the top 10 dominant bacterial genera are *Allobaculum*, *Lactobacillus*, *Bifidobacterium*, *Desulfovibrio*, *Streptococcus*, *Oscillospira*, *Cupriavidus*, *Adlercreutzia*, *Ruminococcus* (Lachnospiraceae), and *Ochrobactrum* (Figure 6A). The number of phylum and genera in the hindgut and fecal microbiota is lower than that of the foregut (Figure 6B). The relative abundance of Firmicutes is highest in the ileum and gradually decreases thereafter (*p* < 0.001); the relative abundance of colon and fecal Bacteroides is significantly higher than that of the duodenum, ileum, and cecum (*p* < 0.001); the Firmicutes/Bacteroides ratio of the colon and fecal segment is significantly lower than that of the duodenum, ileum, and cecum (*p* < 0.001) (Figure 6C). The relative abundance of Proteobacteria is highest in the cecum (*p* = 0.007) (Figure 6C). When compared with the HF-C group, the 4% and 6% FF groups had a lower relative abundance of Proteobacteria (*p* = 0.020) and a higher relative abundance of Actinobacteria in feces (*p* = 0.041); the 6% FF group had a lower relative abundance of Firmicutes in the colon (*p* = 0.041) and feces (*p* = 0.003) (Figure 6C). The results indicated that FF mainly changed the microbial composition of the hindguts at phylum level.

The cluster heat map results of the top 50 abundance genera showed that the microbial composition in cecum, colon, and feces was similar and significantly differed from that in the duodenum and ileum (Figure 7A). LEfSe analysis showed the differential bacterial genera among the intestinal segments in different groups, and the cluster heat map results showed that the difference in intestinal segments (rather than groups) was the main source of differential genera; the microbial composition of the duodenum and ileum were different to those of the colon and feces, and the latter was different from those of the cecum (Figure 7B).

The analysis results of differential bacteria in adjacent intestinal segments are shown in Figure 8. In the LF-C group, there was a relative abundance of the carbohydrate-degrading bacteria, Ruminococcus (Lachnospiraceae) (*p* = 0.006); Oscillospira in the cecum was greater than in the ileum (*p* = 0.005), while the relative abundance of the protein-degrading bacteria, Peptoniphilus (*p* = 0.038) and Leuconostoc (*p* = 0.020), was significantly lower than those in the ileum. The relative abundance of Bacteroides (*p* = 0.044) and Parabacteroides (*p* = 0.008) in the cecum was significantly lower than that in the colon, while the relative abundance of the aerobic bacteria, Salinicoccus (*p* = 0.035), Bacillus (*p* = 0.038), and Yaniella (*p* = 0.030), in the feces was significantly higher (Figure 8A). These microbial changes among the different intestinal segments indistinctly echo the intestinal functions. In the HF-C group, the relative abundances of Akkermansia (*p* = 0.010), Lactobacillus (*p* < 0.001), and Pseudomonas (*p* = 0.040) in the ileum were lower than those in the duodenum. Allobaculum (*p* < 0.001) and Jeotgalicoccus (*p* = 0.043) are enriched in the ileum, and the relative abundance is significantly higher than in the duodenum. The relative abundance of Corynebacterium (*p* = 0.045), Propionibacterium (*p* = 0.034), Anoxybacillus (*p* = 0.003), Staphylococcus (*p* = 0.002), Streptococcus (*p* = 0.005), and Ochrobactrum (*p =* 0.045) in the ileum is significantly higher than in the cecum (Figure 8B). In the 6% FF group, Mucispirillum (*p* = 0.033), Prevotella (*p* = 0.011), Anaerostipes (*p* = 0.007), Oscillospira (*p* < 0.001), Ruminococcus (*p* = 0.001), Bacteroides (*p* = 0.015), Coprococcus (*p* < 0.001), Ruminococcus (Lachnospiraceae) (*p* = 0.003), and Allobaculum (*p* = 0.019) are enriched in the cecum, and the relative abundance is significantly higher than that of the ileum (Figure 8D). The above results indicated that high-fat diets caused significant changes in the microbial composition of the foregut, while functional fiber supplementation promoted an increase in the abundance of bacteria related to carbohydrate metabolism in the hindgut.

### 2.4. Microbial Traceability Analysis

To explore the microbial replenishment process, the anterior intestinal segment samples served as potential sources to predict the origins of microbial OTUs in the posterior intestinal segment. Compared with the LF-C group, the microbial transfer from duodenum to the posterior intestinal segments decreased in the HF-C group (Figure 9A,B). Unknown factors, such as the intestinal environment and the backwash of posterior intestinal segment microbes, played more important roles in the origin of microbial OTUs in the ileum and cecum of the HF-C group than in those of the LF-C group. The microbial transfer from cecum to colon and feces in the HF-C group increased compared to that in the LF-C group. The 4% FF group had a similar traceability analysis result to the LF-C group (Figure 9C). The microbial transfer from the duodenum to the posterior intestinal segments in the 6% FF group increased more than in the other groups (Figure 9D). These results indicated that the multiple regulatory mechanisms of dietary nutrients on the microbiota composition existed, not only in relation to the microbial metabolism, but also to the microbial migration.

### 2.5. Changes in Microbial Metabolite Short-Chain Fatty Acids (SCFAs)

Except for the duodenum (due to insufficient samples), we determined the levels of SCFAs in the contents of the ileum, cecum, colon, and feces, as shown in Figure 10. The results showed that the caecal levels of propionate (*p* = 0.023), butyrate (*p* = 0.010), and the total SCFAs (*p* = 0.025) in the HF-C group decreased compared to those in the LF-C group. Compared to the HF-C group, the 4% and 6% FF groups had greater levels of ileal acetate (*p* = 0.033) and total SCFAs (*p* = 0.023), and caecal acetate (*p* = 0.010), butyrate (*p* = 0.035), and total SCFAs (*p* = 0.023), as well as colonic acetate (*p* = 0.047). Furthermore, the 6% FF group had decreased levels of colonic isovalerate (*p* = 0.010) and isobutyrate (*p* = 0.013), and increased levels of total fecal SCFAs (*p* = 0.014) compared to the HF-C group. These results indicated that the microbial degradation of FF began in the foregut and was maximal in the hindgut.

## 3. Discussion

The gastrointestinal tract is a multiorgan system with great regional diversity, housing extensive gut microbes and providing diverse functions. The difference in the environment and function of different intestines affects the colonization of microbiota [2]. In this study, we found that the structure of gut microbiota in the foregut and the hindgut are significantly different; the species diversity of the foregut microbiota is significantly lower than that of the hindgut. Compared to the foregut, the abundance of Firmicutes in the hindgut was decreased, and the abundance of Bacteroides phylum was increased. Bacteria are not uniformly distributed throughout the gut lumen, and the microbial abundance and diversity of vertebrates along the entire gut usually increases from proximal to distal [1]. The short transit time and excretion of digestive enzymes and bile in the foregut, which may not be conducive to gut microbiota colonization [13], may be the reason for the higher relative abundance of Firmicutes in the foregut [14]. The gut environment could influence bacterial metabolism and competition; the hindgut is colonized with a large number of microorganisms, interacting with each other to maintain intestinal homeostasis [15]. The increased Bacteroides phylum in the hindgut may be related to a stronger ability to ferment complex polysaccharides in the diet [16], which may create opportunities for host health through the degrading of energy-rich complex carbohydrates [15].

In this study, the high-fat diet significantly changed the microbial composition (evidenced by increased unique OTUs and β diversity) in different intestinal segments of mice, especially in the foregut (duodenum and ileum), which is similar to the findings of Martinez-Guryn et al. (2018) [17]. The traceability analysis indicated that a high-fat diet decreased the microbial transfer from the duodenum to the posterior intestinal segments, and increased the influence of unknown factors on the microbial composition in the ileum and cecum. Microbes gradually migrated backward with intestinal peristalsis, and colonized after adapting to the intestinal environment. Studies have shown that high-fat diets can promote the secretion of bile in the intestinal tract [18], thereby increasing the difficulty of microbial colonization in the foregut segment and inhibiting bile-intolerant bacteria [5]; this was highlighted by the increased bile-resistant bacterium Jeotgalicoccus and the many decreased bacteria in the HF-C group in this study. On the contrary, the supplementation of 6% functional fiber increased the microbial transfer from the duodenum to the posterior intestinal segments, and increased the common OTU number in the whole intestine. Functional fibers might be helpful for the microbial transfer attributed to the increased intestinal content viscosity and intestinal retention time [19]. This result indicates that it might be useful to promote the microbial colonization, such as probiotics, in the hindgut by adding functional fibers.

In this study, the influence of functional fiber on gut microbiota mainly occurred in the hindgut. The most direct evidence was that 6% functional fiber supplementation significantly reduced the relative abundances of Firmicutes and Proteobacteria, and increased the relative abundance of Bacteroidetes in the hindgut. At the genus level, 6% functional fiber significantly promoted the bloom of carbohydrate metabolism-related bacteria, such as Mucispirillum, Prevotella, Oscillospira, Ruminococcus, and Bacteroides, in the cecum compared to that in the ileum. Concurrently, the functional fiber significantly promoted the production of SCFAs in the cecum and the colon, especially acetate and butyrate, both of which are beneficial to intestinal health and the reduction of inflammatory factors (lipopolysaccharide) from microbes. Dietary fiber is the residue of plant food resistant to hydrolysis by human alimentary enzymes [20]. A large number of microbes colonize in the hindgut, which encode a variety of carbohydrate-active enzymes, such as glycosidase, glycosyltransferase, polysaccharide lyase, and carbohydrate esterase [21]. Therefore, functional fiber may induce a change in intestinal microbiota composition by acting as a substrate. Furthermore, the consumption of dietary fiber accords with the metabolic characteristic of intestinal microbiota, which may be an important basis for the symbiotic relationship between intestinal microbiota and its host.

## 4. Materials and Methods

### 4.1. Animal Diets

The composition of all purified diets used in this study are listed in Appendix A. The energy content of the low-fat basal diet (LFD) was 3.6 kcal/g, with 19% kilocalories (kcals) from protein, 71% from carbohydrates, and 10% from fat. The calorific value of the high-fat basal diet (HFD) was 5 kcal/g, with 19.4% of the calories from protein, 20.6% from carbohydrate, and 60% from fat. The normal chow diet (NCD) derived 14.1% of its calories from protein, 75.9% from carbohydrates, and 10% from fat, for a total energy content of 3.6 kcal/g. All feeds were purchased from Trophic Animal Feed High-tech Co., Ltd. (nan tong, Jiangsu, China). The basal feed was supplemented with a suitable amount of FF (14.3% guar gum (Yunzhou Science and Technology Corp., Ltd., Yunzhou, Shangdong, China) and 85.7% pregelatinized waxy maize starch (Puluoxiang Starch Corp., Ltd., Hangzhou, Zhejiang, China)) to replace cellulose.

### 4.2. Treatment

Specific pathogen-free 4- to 5-week-old male C57BL/6 mice were purchased from the Laboratory Animal Center, Huazhong Agricultural University, Wuhan, People’s Republic of China, and housed at 22–24 °C under a 12 h light/dark diurnal cycle with food and water provided ad libitum. After a 7-day adaptation period, the mice were fed on a low-fat diet (LFD) or a high-fat diet (HFD). Ten weeks later, the mice were weighed; obese mice were identified as those with at least a 20% weight gain compared to the LFD-fed mice. Twenty-one HFD-induced obese (DIO) mice were randomly divided into 3 groups (*n* = 7/group) and were fed on a HFD supplemented with 0%, 4%, and 6% FF for 12 weeks (period 1); they were then fed on an NCD supplemented with 0%, 4%, and 6% FF for 4 weeks (period 2). They were referred to as the HF-C group, 4% FF group and 6% FF group, respectively. Seven LFD-fed lean mice (referred to as the LF-C group) were still fed on an LFD in period 1 and an NCD in period 2, as a normal control group. Food intake was measured daily, and body weight was measured weekly. An overview of the study design is shown in Appendix A.

### 4.3. Sample Collection

At the end of the experiment, the mice were dissected to collect the contents of the duodenum, ileum, cecum, and colon, and a fecal sample was also collected. All samples were collected and stored in a refrigerator at −80 °C.

### 4.4. DNA Extraction, 16S rDNA Amplification, and Illumina MiSeq Sequencing

Microbial DNA was extracted from 220 mg of each mouse’s intestinal content and fecal sample with a QIAamp Fast DNA Stool Mini Kit (Qiagen, Germany), according to the manufacturer’s instructions. Successful DNA isolation was achieved by separation using agarose gel electrophoresis. The forward primer 341F (50-ACTCCTACGGGAGGCAGCAG-30) and the reverse primer 806R (50-GGACTACHVGGGTWTCTAAT-30) were used for the amplification of the V3–V4 hypervariable region of a 16S rRNA gene. The PCR conditions were a predenaturation cycle at 94 °C for 4 min, 25 cycles of denaturation at 94 °C for 30 s, annealing at 50 °C for 45 s, elongation at 72 °C for 30 s, and a final postelongation cycle at 72 °C for 5 min. The PCR products were purified with AmpureXP beads (AGENCOURT). After purification, the PCR products were used for the construction of the libraries and then paired-end sequenced (2 × 250) on a MiSeq platform (Illumina, San Diego, CA, United States) at the Beijing Genomics Institute (Beijing, China).

### 4.5. Sequence Filtering, OTU Clustering, and Sequence Analyses

Sequences with an average Phred score of less than 30, ambiguous bases, homopolymer runs exceeding 6 bp, primer mismatches, and sequence lengths shorter than 100 bp were removed. Only sequences with overlaps longer than 10 bp and without any mismatches were assembled according to their overlap sequences. Reads that failed to assemble were discarded. Barcode and sequencing primers were trimmed from the assembled sequence. To avoid the effect of sequencing depth on the composition of microbiota (Hughes and Hellmann, 2005), we rarefied the library size to 27871 tag-depth per sample by using the rarefy function in R package. High-quality tags were clustered into operational taxonomic units (OTUs) at 97% similarity with USEARCH (Edgar, 2010). Taxonomy assignments for the 16S rRNA gene sequences were performed with the RDP Classifier program (V2.2) (Wang et al., 2007) and the SILVA 16S sequence database. A Venn diagram was generated for comparison among the OTUs of the groups. The alpha diversity values of each sample were assessed on the basis of the observed OTUs and bias-corrected using the Chao richness estimator (Chao 1). Beta diversity measures depended on unweighted UniFrac distance and were calculated using mothur. Metastats was used to identify bacterial taxa differentially represented between distinct intestinal segments at genus or higher taxonomy levels.

### 4.6. Short-Chain Fatty Acids Determination

The levels of acetate, propionate, butyrate, isobutyrate, valerate, and isovalerate in the feces and intestinal contents were measured by gas chromatography. Briefly, approximately 10–30 mg of samples was homogenized in 500 μL methanol, ground to a fine powder using a grinding mill at 65 HZ for 120 s, and then vortexed for 30 s. The samples were centrifuged at 12,000 rpm and 4 °C for 15 min, and a 400 μL supernatant from each sample was concentrated by centrifuging once more. Each sample was redissolved in 50 μL methanol and centrifuged at 12,000 rpm and 4 °C for 15 min. One microliter of each sample was subjected to gas chromatography (GC 2010, Shimadzu, Japan, equipped with a CP-Wax 52 CB column of 30.0 m × 0.53 mm i.d., Chrompack, Middelburg, The Netherlands). SCFAs were quantified using standard curves of 0.5 to 100 μM organic acids (Fluka, Buchs, Switzerland).

### 4.7. Statistical Analyses

The ANOVA repeated measures program of SAS software (Statistical Analysis System 9.4, SAS institute, Cary, NC, USA) was used for statistical analyses. When the microbial abundance ratio does not fall within normal distribution and the variance is not uniform, the Mann–Whitney U test is used, and the error rate is corrected by Bonferroni. Metastats analysis combined with the Benjamini–Hochberg correction method was used to compare the differences of intestinal microbes at different classification levels. The RStudio 0.97.310 program package in R3.0.2 was used for Spearman’s correlation analysis, and the results were corrected using Benjamini–Hochberg for FDR. When *p* < 0.05, the difference is significant.

## 5. Conclusions

In conclusion, the response of microbiota from different intestinal segments to diet is related to the intestinal physiologic function and the physicochemical properties of dietary nutrients, as well as the microbial spatial specificity. Specifically, a high-fat diet hinders the microbial transfer and colonization from the foregut to the hindgut, possibly by causing huge changes in the digestive environment of the foregut, and then increasing the regional heterogeneity of microbial composition in the whole intestine. Functional fiber, with its unique physicochemical properties, especially viscosity and fermentability, promotes the microbial transfer and colonization from the foregut to the hindgut, and facilitates the development of microbiota in the hindgut towards to the utilization of fiber fermentation. Knowledge of these links can help us to select the right diet to target the adjustment of intestinal microbes.

## Figures and Tables

**Figure 1 ijms-22-06525-f001:**
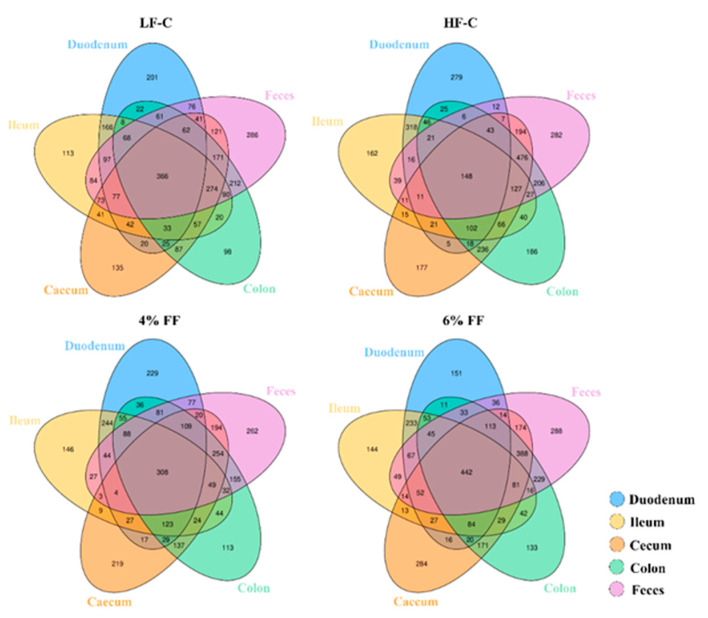
Venn diagram of common and unique OTUs in intestinal microbiota in different intestinal contents and feces.

**Figure 2 ijms-22-06525-f002:**
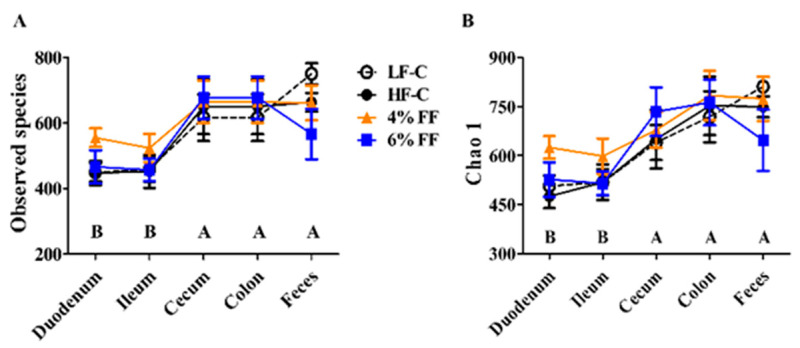
Changes in microbial α diversity in different intestinal segments: (**A**) the number of observed OTUs, and (**B**) the bias-corrected Chao richness estimator (Chao 1). Data are represented as mean ± S.E.M., *n* = 5–7. Significance is considered at *p* < 0.05. AB means in the same bar without a common letter differ at *p* < 0.01.

**Figure 3 ijms-22-06525-f003:**
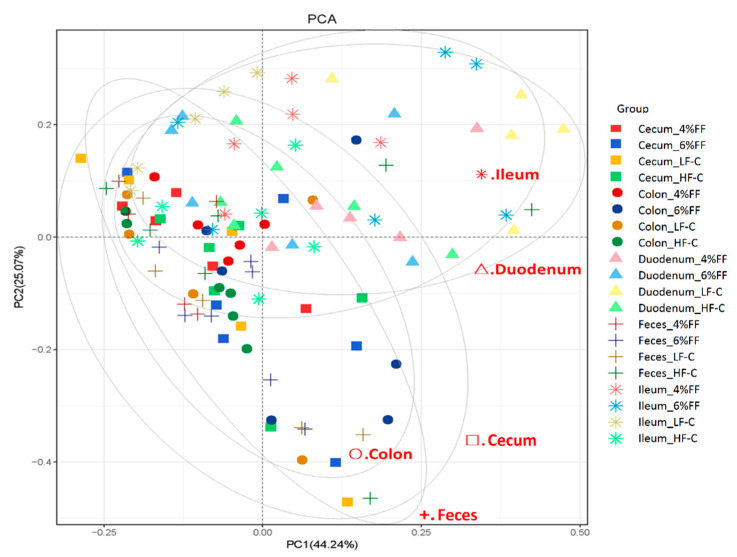
Changes in microbial β diversity in different intestinal segments (*n* = 5–7).

**Figure 4 ijms-22-06525-f004:**
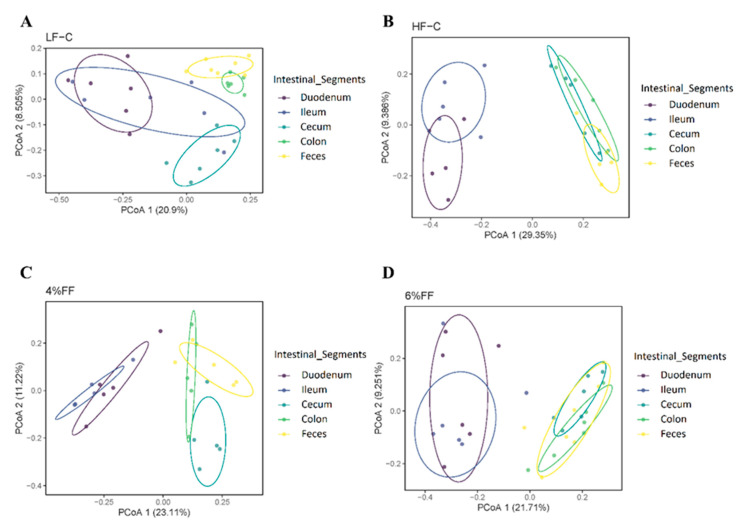
Principal coordinate analysis of different intestinal segments in each group (*n* = 5–7): (**A**) LF-C, (**B**) HF-C, (**C**) 4% FF, and (**D**) 6% FF.

**Figure 5 ijms-22-06525-f005:**
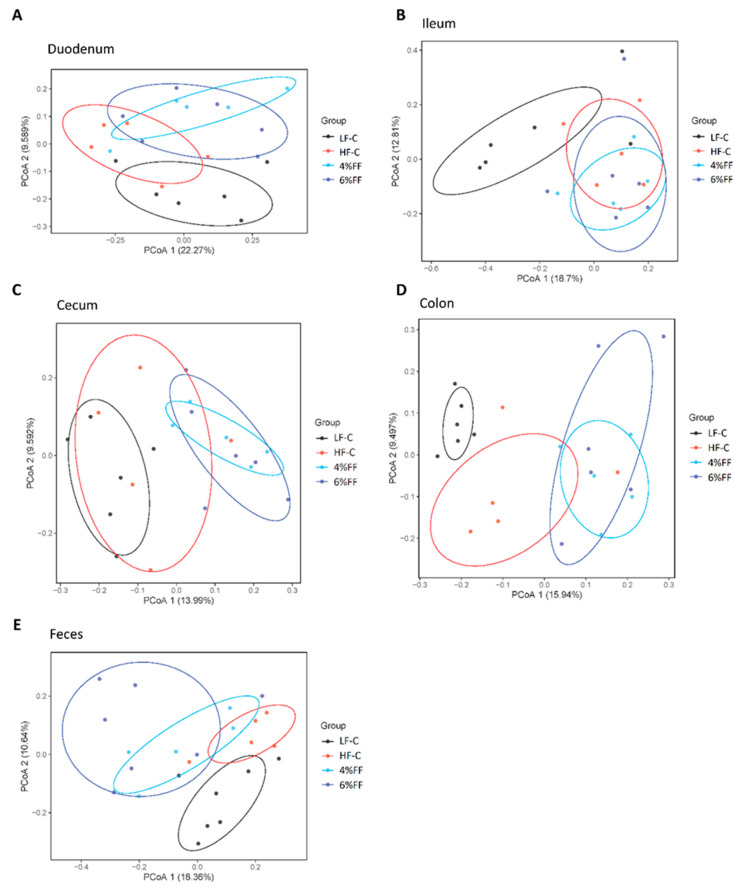
Principal coordinate analysis of different groups in each intestinal segment (*n* = 5–7): (**A**) Duodenum, (**B**) Ileum, (**C**) Cecum, (**D**) Colon, and (**E**) Feces.

**Figure 6 ijms-22-06525-f006:**
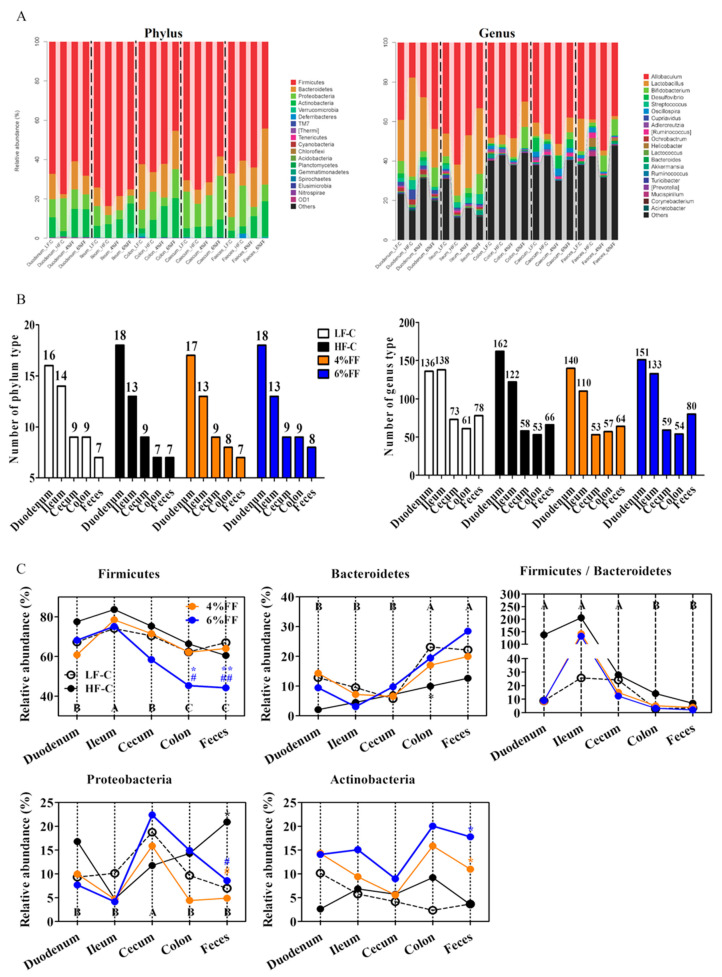
Microbiota composition at phylum and genus levels: (**A**) taxonomic distribution of bacterial phyla and genera, (**B**) numbers of phyla and genera in microbial composition, and (**C**) changes in relative abundance of the most dominant phyla (*Firmicutes*, *Bacteroidetes*, *Proteobacteria*, *Actinobacteria*) and *Firmicutes*/*Bacteroidetes* ratio.

**Figure 7 ijms-22-06525-f007:**
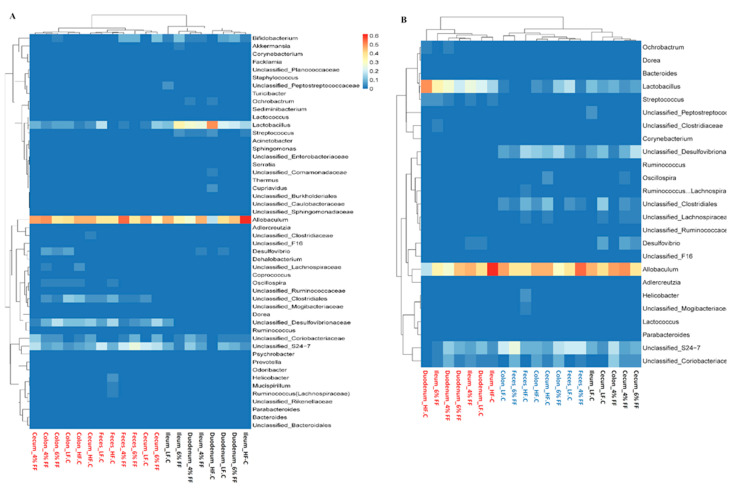
Microbial composition and changes in different intestinal segments: (**A**) clustering heat map of the first 50 genera, and (**B**) clustering of differential genera among intestinal segments in different groups by LEfSe analysis.

**Figure 8 ijms-22-06525-f008:**
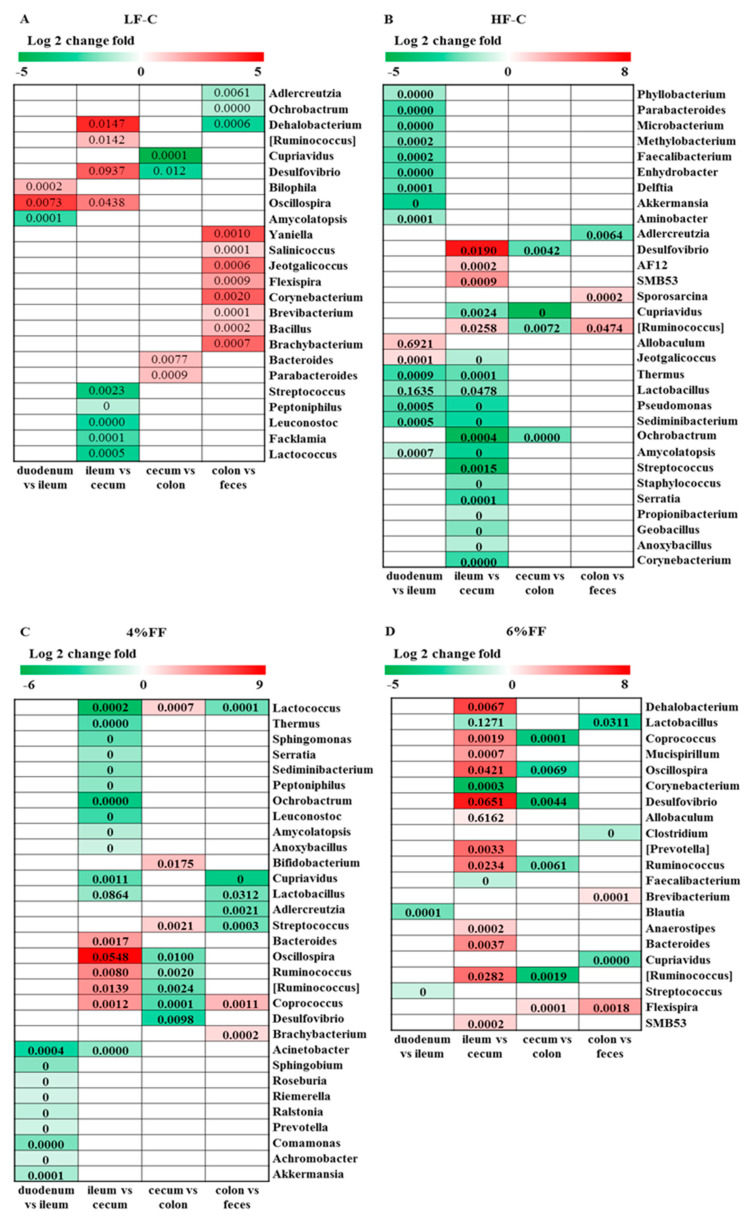
The differential genera between adjacent intestinal segments in each group: (**A**) LF-C, (**B**) HF-C, (**C**) 4% FF, and (**D**) 6% FF.

**Figure 9 ijms-22-06525-f009:**
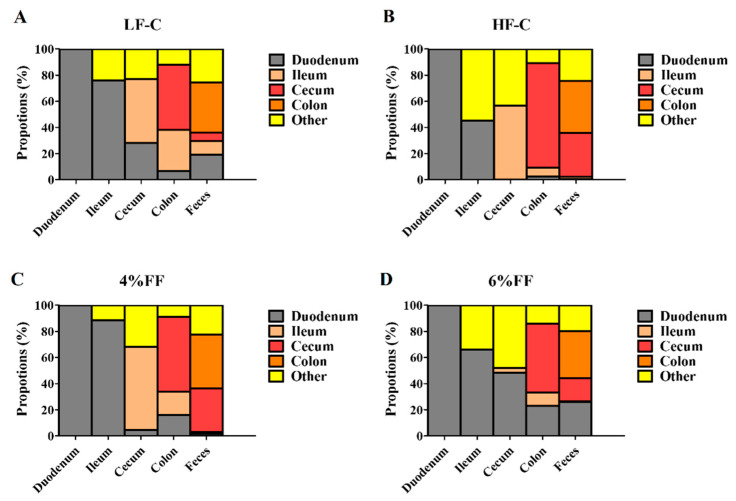
Tracing the source of OTUs in each intestinal segment (**A**–**D**). The anterior intestinal segment samples were taken as potential sources of the latter intestinal segment. The chart indicates the top 100 most abundant OTUs.

**Figure 10 ijms-22-06525-f010:**
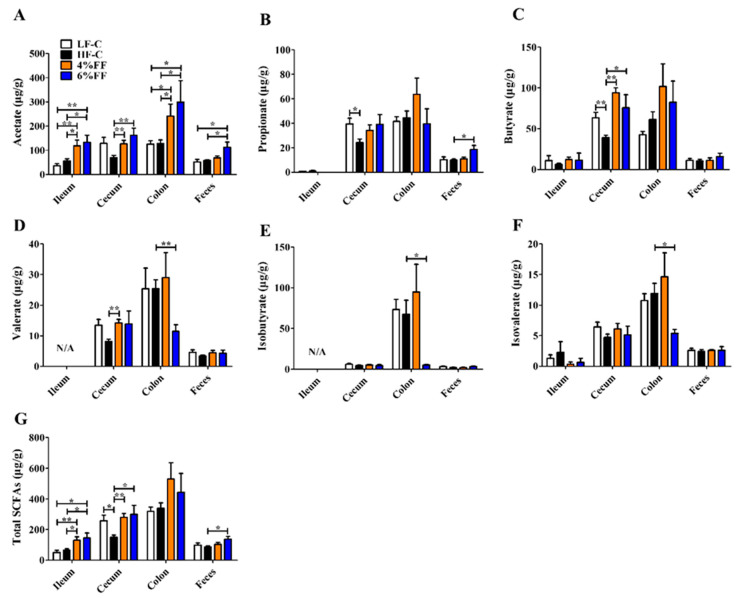
Changes in the production of short-chain fatty acids (SCFAs) in each intestinal segment: acetate (**A**), propionate (**B**), butyrate (**C**), valerate (**D**), isobutyrate (**E**), isovalerate (**F**), and total SCFAs (**G**). Data are represented as mean ± S.E.M., *n* = 5–7. Significance is considered at *p* < 0.05. * means in the same bar without a common letter differ at *p* < 0.05; ** means in the same bar without a common letter differ at *p* < 0.01.

## Data Availability

Not applicable.

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
