# Peer review of "The Effect of Functional Fiber on Microbiota Composition in Different Intestinal Segments of Obese Mice"

_ijms, 2021, doi:10.3390/ijms22126525_

Round 1

Reviewer 1 Report

The authors have investigated the effect of a high-fat vs. low-fat diet on the microbial composition of different parts of the intestinal tract.  They also investigated the effect of fiber supplementation on the microbial composition.

Major points

  1. How were the composition of the diets determined? More specifically, why did the low-fat diet and the normal-chow diet have the same percentage of kcals from fat?

  1. The authors should make a flow diagram displaying how the mice were grouped after 3 weeks, and the different diets they were given afterward. That will make the groupings easier for the readers to understand.

  1. The authors have shown that A) the beta diversity increases distally along the intestinal tract and B) a high-fat diet increases further the beta diversity. But what is the potential clinical significance of these changes?  There needs to be more emphasis in the discussion section about the ramifications or translational importance of these findings.

  1. One potential bias in this study is that there may be a correlation between mouse weight and microbial composition. Did the authors weigh the mice at the end of the study prior to specimen collection?  In the methods section, it says that the mice were weighed weekly.  But did mice in the low-fat group who had increased weight display a similar microbial composition to the high-fat mice or the obese mice?

  1. There are some minor grammatical errors. As an example, in line 24 the authors say that a high-fat diet “drived” instead of “drove.”  Either a review by the authors, or use of an English translation service, would be helpful.  But the errors are not major or prevalent.

Minor points

  1. In lines 42-43, the authors say that the “microbial profiles of the fore-, mid- and hind-intestine segments show a gradual increasing trend. Do the authors mean that there is a higher absolute abundance of organisms?

Reviewer 2 Report

The manuscript was reviewed for publication in the journal. The manuscript was designed to evaluate the regulatory effect of functional fiber on microbiota composition in different intestinal segments of obese mice. It is the reviewer’s opinion that the manuscript is easy to follow. However, it appears that there are a couple of concerns in the manuscript.

1) The authors have discussed, but the importance of the study regarding the effects of different diets on microbiota from different intestinal segments appears to be questionable after reading the manuscript.

2) The authors showed that the microbiota compositions in different intestinal segments were changed among the groups with different diets and functional fiber by analyzing the methods of the observed OTUs, a diversity, b diversity, principal coordinate analysis, microbiota composition, and cluster heat map. After all, it appears that the authors performed the various types of microbiota analysis. In addition, the authors did not discuss the influence of different microbiota compositions to mice.

3) The authors did not mention the details of functional fiber (FF) and the reason to use this FF. What is CSF (Page 3, Line 77)? 4% and 6% FF were used in the study. Is there any reason of these concentrations? The authors should explain the issue.

4) Figure 10 showed the changes in the production of SCFAs in each different intestinal segment. How about the correlation between the changes of SCFA and microbiota compositions? The authors should discuss the issue.

5) Minors:

Page 13, Line 281: segmetns

The name of microbiota should be italicized?

Author Response

RE: The effect of functional fiber on microbiota composition in different intestinal segments of obese mice

Dear reviewer,

Thank you very much for your warm composition comments concerning our manuscript entitled " The effect of functional fiber on microbiota composition in different intestinal segments of obese mice" (MS ID: IJMS-1216587).

Here we submit a new version of our manuscript, which has been modified accordingly to the comments. Detailed point-by-point responses are provided below. We mark all the changes by using red font in the revised manuscript.

We want to do our best to meet the requirements. If you have any question about this paper, please don’t hesitate to let us know.

Sincerely,
Rong Zheng and Chuanhui Xu

Comment 1: The authors have discussed, but the importance of the study regarding the effects of different diets on microbiota from different intestinal segments appears to be questionable after reading the manuscript.

Response: Thanks for your comment. In the study, we found there was spatial difference in the microbiota composition of different intestinal segments. Traceability analysis showed different dietary components might influence the spatial difference in different ways, such as dietary fat effect the bile secretion in small intestine, while functional fiber act as an available fermentation substrate for the large intestinal microflora. These results suggested a complex link between the diet and intestinal microbiota in the levels of physicochemical property of dietary components, physiological function of intestinal segments and microbial spatial specificity. In addition, functional fibers promote the transfer of anterior intestinal microbes to the posterior intestinal segment, which might be related to its high viscosity. Clinically, functional fibers may help probiotics colonizng in the gut. These expressions were appropriately enriched in discussion and conclusion of the revised version.

Comment 2: The authors showed that the microbiota compositions in different intestinal segments were changed among the groups with different diets and functional fiber by analyzing the methods of the observed OTUs, a diversity, b diversity, principal coordinate analysis, microbiota composition, and cluster heat map. After all, it appears that the authors performed the various types of microbiota analysis. In addition, the authors did not discuss the influence of different microbiota compositions to mice.

Response: Thanks for your comment. In previous study, we found that the addition of FF to a continuous high fat diet and normal chow diet can effectively improve insulin sensitivity in obese mice, which was significantly related with the changes in fecal microbial composition [1]. Based on this, this study tracked the changes of intestinal microbiota arong the axis of the gastrointestinal tract in different diets, aiming to illustrate how dietary differences cause the differences of intestinal microbiota. Specifically, this study compared the microbiota composition in different intestinal segmetns within the same group or the different gorups within the same intestinal
segment, including α-diversity, β-diversity and the compositions at phylum and genus levels. Furthermore, the traceability analysis explored the connections among the microbiota composition of intestinal segments under different diets.
As for the impact of different microbiota composition on mice, by using procrustes analysis, we found that the colonic microbiota composition had a significant correlation with the mice phenotypes, including weight and insulin sensitivity indicators, compared to the that of other intestinal segments, which suggested that there may have been a significant impact on the host phenotype by the regulation of colon microorganisms. However, since the phenotypic data have been reported, this part of the results were not deeply discussed in this study.

Comment 3: The authors did not mention the details of functional fiber (FF) and the reason to use this FF. What is CSF (Page 3, Line 77)? 4% and 6% FF were used in the study. Is there any reason of these concentrations? The authors should explain the issue.

Response: Thanks for your comments. FF was created by our research group, its components was 14.3% guar gum and 85.7% pregelatinized waxy maize starch (lines 78-81 of revised manuscript). Rodent study show that the successive supplementations of 4% and 6% FF in high-fat diet and normal chow diet improve the obesity-related metabolic disorders, and regulate the fecal microbiota composition and the productions of SCFAs with a dose-dependent manner [1]. However, it is still unclear how FF affects the microbial composition and metabolism in the gut. Therefore, this study tracked the changes of intestinal microbiota arong the axis of the gastrointestinal tract in different diets, aiming to illustrate how dietary differences cause the differences of intestinal microbiota. The relevant informations see lines 60-67 of the revised manuscript.
“CSF” is a writing error and we have corrected “FF” (line 78 of the revised version).

Comment 4: Figure 10 showed the changes in the production of SCFAs in each different intestinal segment. How about the correlation between the changes of SCFA and microbiota compositions? The authors should discuss the issue.

Response: Thanks for your comments. SCFAs are important products of intestinal microbial metabolism. Acetate, propionate and butyrate are the main metabolites of intestinal microbes using undigested dietary carbohydrate or host-derived glycans as substrates, while isobutyrate and isovalerate are the metabolites of microbes using protein and peptide as substrates [2,3]. Therefore, the yield of SCFAs reflect microbial metabolic activity in gut, which is closely related to the microbial composition. This study found that, after switching to a uniform nornal chow diet, obese individuals had similar levels of SCFAs in each intestinal segment from thin normal individuals, but a significant increase existed in obese mice supplemented with FF. Specifically, FF supplementation increased the levels of acetate in ileum, cecum and colon and butyrate in cecum in a dose-dependent manner, 6% FF supplementation reduced the levels of colonic isobutyrate and isovalerate. Acetate and butyrate are the key active substances for dietary fiber to improve the intestinal health and host metabolism [4]. On the microbiota composition, FF increased the relative abundance of Bacteroidetes capable of utilizing diet fiber, and reduced the relative abundance of Firmicutes capable of adapting to the high bile condition induced by high fat diet. Thus, FF affects the microbial metabolic activity and also affects the changes of the microbial composition. We have enriched the relevant expressions in the discussion, see lines 320-335 of the revised version. Thanks again for your valuable advice.

Comment 5: Minors: Page 13, Line 281: segmetns. The name of microbiota should be italicized?

Response: Thanks for your comments. We have corrected this spelling error (line 281 of revised manuscript). Furthermore, we have carefully checked throughout the manuscript to reduce errors. According to “the international code of nomencluture of bacteria”, "genus" above grades, like "order" and "family", written in orthographic, italic for the genus name and species name. Thanks again for your valuable advice.

Reference
1. Xu, C.; Liu, J.; Gao, J.; Wu, X.; Cui, C.; Wei, H.; Zheng, R.; Peng, J., Combined Soluble Fiber-Mediated Intestinal Microbiota Improve Insulin Sensitivity of Obese Mice. Nutrients 2020, 12, (2).
2. Topping, D.L.; M., C.P. Short-chain fatty acids and human colonic function: Roles of resistant starch and nonstarch polysaccharides. Physiol. Rev. 2001, 81, 1031–1064, doi:10.1152/physrev.2001.81.3.1031.
3. Desai, M.S.; Seekatz, A.M.; Koropatkin, N.M.; Kamada, N.; Hickey, C.A.; Wolter, M.; Pudlo, N.A.; Kitamoto, S.; Terrapon, N.; Muller, A. A dietary fiber-deprived gut microbiota degrades the colonic mucus barrier and enhances pathogen susceptibility. Cell 2016, 167, 1339–1353.e1321, doi:10.1016/j.cell.2016.10.043.
4. Suzuki, T.; Yoshida, S.; Hara, H. Physiological concentrations of short-chain fatty acids immediately suppress colonic epithelial permeability. Br. J. Nutr. 2008, 100, 297–305, doi:10.1017/S0007114508888733.

Round 2

Reviewer 2 Report

The manuscript was re-reviewed for publication in the journal. The manuscript was designed to evaluate the regulatory effect of functional fiber on microbiota composition in different intestinal segments of obese mice. It is the reviewer’s opinion that the revised manuscript appears to become interesting. The authors promptly explained/discussed all issues suggested. I have no more concern in the manuscript.